# DNA Sequence-Dependent Properties of Nucleosome Positioning in Regions of Distinct Chromatin States in Mouse Embryonic Stem Cells

**DOI:** 10.3390/ijms232214488

**Published:** 2022-11-21

**Authors:** Guoqing Liu, Zhi Zhang, Biyu Dong, Jia Liu

**Affiliations:** 1School of Life Science and Technology, Inner Mongolia University of Science and Technology, Baotou 014010, China; 2Inner Mongolia Key Laboratory of Functional Genomics and Bioinformatics, Inner Mongolia University of Science and Technology, Baotou 014010, China; 3School of Science, Inner Mongolia University of Science and Technology, Baotou 014010, China

**Keywords:** nucleosome occupancy, DNA bending energy, rotational positioning, chromatin state

## Abstract

Chromatin architecture is orchestrated, and plays crucial roles during the developmental process by regulating gene expression. In embryonic stem cells (ESCs), three types of chromatin states, including active, repressive and poised states, were previously identified and characterized with specific chromatin modification marks and different transcription activity, but it is largely unknown how nucleosomes are organized in these chromatin states. In this study, by using a DNA deformation energy model, we investigated the sequence-dependent nucleosome organization within the chromatin states in mouse ESCs. The results revealed that: (1) compared with poised genes, active genes are characterized with a higher level of nucleosome occupancy around their transcription start sites (TSS) and transcription termination sites (TTS), and both types of genes do not have a nucleosome-depleted region at their TTS, contrasting with the MNase-seq based result; (2) based on our previous DNA bending energy model, we developed an improved model capable of predicting both rotational positioning and nucleosome occupancy determined by a chemical mapping approach; (3) DNA bending-energy-based analyses demonstrated that the fragile nucleosomes positioned at both gene ends could be explained largely by enhanced rotational positioning signals encoded in DNA, but nucleosome phasing around the TSS of active genes was not determined by sequence preference; (4) the nucleosome occupancy landscape around the binding sites of some developmentally important transcription factors known to bind with different chromatin contexts, was also successfully predicted; (5) the difference of nucleosome occupancy around the TSS between CpG-rich and CpG-poor promoters was partly captured by our sequence-dependent model. Taken together, by developing an improved deformation-energy-based model, we revealed some sequence-dependent properties of the nucleosome arrangements in regions of distinct chromatin states in mouse ESCs.

## 1. Introduction

The hierarchically compacted structure of chromatin underlies the functions of eukaryotic cells, by regulating fundamental molecular processes such as transcription, DNA replication and repair, and recombination. A nucleosome is the basic unit of chromatin structure, and, therefore, plays extremely important roles in establishing high-order chromatin structure, regulating gene transcription, and consequently modulating cell functions during embryo development and disease-relevant processes. For example, interactions between spatially adjacent nucleosomes and nucleosome spacing contribute to the formation and diversity of 10–30 nm chromatin fiber [1,2,3,4]. H1 linker histone and epigenetic marks such as H3K27 and H3K36 methylation impact local chromatin compaction [5]. High-throughput sequencing techniques such as Hi-CO and Micro-C enable the exploration of genome-wide nucleosome folding and their connection with gene regulatory mechanisms [6,7,8]. In the aspect of nucleosome-mediated regulatory roles, nucleosome-depleted regions are known to assist the binding of regulatory proteins such as transcription factors to target sites on the DNA template. Nucleosomes may also contribute to RNA Pol II pausing, thereby regulating transcription [9,10].

Regarding the nucleosome position, a traditionally popular mapping approach is MNase-seq, which digests chromatin with micrococcal nuclease (MNase) followed by high-throughput sequencing of obtained nucleosome-sized fragments and mapping. MNase-seq has greatly expanded chromatin biology. However, it was demonstrated that MNase-seq caused inaccurate mapping of nucleosome positions due to sequence bias in MNase cleavage [11,12]. This type of error was shown to be non-negligible when chromatin was digested with variable MNase concentration [13,14]. Mutants introduced at specific sites of histones (H3 or H4) enabled Fenton reactions at sites when copper ions and hydrogen peroxide were present, and hydroxyl radicals generated by Fenton reactions caused site-directed DNA cleavage near nucleosome dyads [14,15]. Subsequent sequencing and computational methods were used to determine nucleosome centers with base-pair resolution. This kind of approach, termed chemical mapping, was applied to yeast and mouse [10,14,15], discovering some novel nucleosome positioning patterns that were not detectable by MNase-seq. One of the most remarkable findings from the comparative analysis between chemical mapping and MNase-seq mapping was the enrichment of MNase-sensitive (fragile) nucleosomes near the TSS and TTS in mouse ESCs [10]. In addition, computational approaches achieved great progress in predicting nucleosome positions and understanding nucleosome positioning rules [16,17,18,19,20,21,22]. In this field, we developed two biophysical models in which DNA deformation energy was calculated to indicate nucleosome positioning [21,22]. One model used global constraints of nucleosomal DNA structure in the deformation energy calculation [21,23], while the other utilized the local path of nucleosomal DNA [22]. Both models achieved good performance in predicting the MNase-seq-determined nucleosome occupancy and chemical-mapping-indicated rotational positioning of nucleosomes in budding yeast [21,24]. In this study, we employed the deformation energy model [21] and developed a bending-energy-based method for measuring nucleosome occupancy to investigate the sequence-dependent properties of nucleosome positioning in mouse ESCs.

In this study, we focus on the nucleosome positioning in the regions with distinct states in ESCs. Current studies have suggested that chromatin adopts at least three major states in embryonic stem cells [25,26], including ‘active’, ‘repressive’ and ‘poised’ states. The active state is enriched with active histone mark H3K4me3, and underlying genes are highly active in transcription, while the repressive state is characterized with repressive mark H3K27me3 and gene silencing. The poised state carries both the opposing marks of H3K4me3 and H3K27me3, and, thus, is also termed ‘bivalent’ state. The poised state in ESCs was known to silence developmental genes and represent a ‘ready for rapid activation’ state required to maintain the pluripotency of stem cells and induce gene activation once the cell differentiation program was initiated [25,26]. A previous MNase-seq based analysis showed that nucleosomes were depleted at the TSS for mouse ESCs, and the nucleosome depleted region (NDR) extended into the gene body [27]. The genes with a high transcription level were characterized with broader and deeper NDR. At the TTS in ESCs, NDR was detected but largely located at the near upstream of the TTS. When analyzed separately, both active and bivalent promoters were found to be devoid of nucleosomes, whereas the dip of the NDR of repressive promoters attenuated substantially [27]. By contrast, a study based on the chemical mapping approach discovered a positive correlation between nucleosome occupancy and gene transcription level, both upstream and downstream of TSS [10]. This discrepancy was interpreted as the presence of MNase-sensitive nucleosomes that were susceptible to MNase digestion. MNase-sensitive nucleosomes might be reflective of a complicated effect caused by nucleosome accessibility [13], MNase cleavage bias [12], and nucleosome stability mediated by DNA deformability [22]. This is strongly suggestive of the demand to unveil the nature of MNase-sensitive nucleosomes. Accordingly, in this study, we make an effort to model the chemical map of nucleosomes in ESC. As mentioned above, the chromatin state has usually been defined according to histone modification marks, but the local compaction state of nucleosomes remains largely elusive. Titration of MNase with varying concentrations was used to determine nucleosome accessibility [13]. Another approach, called TC-rMNase-seq, was developed to study the local chromatin structure of mouse ESC [28], whereby a time-course digestion of chromatin using moderate MNase levels (TC-rMNase-seq) was performed, and two types of local chromatin structure (e.g., open chromatin regions and moderately condensed regions) were defined. This could be an alternative way to study local chromatin compaction, and the corresponding data were integrated in this study.

Here, we present a computational analysis of distinct chromatin states in mouse embryonic stem cells to address the following questions: (1) What characteristic patterns do the chromatin states, including active, repressed and bivalent states in embryonic stem cells, have, in terms of nucleosome positioning? (2) Are there any sequence-dependent properties underlying the distinct chromatin states?

## 2. Results and Discussion

### 2.1. Nucleosome Organization around Promoters with Distinct Chromatin States in ESCs

Before we explore the sequence-dependent property of chromatin states in ECSs, it is important to see how genes with distinct promoter states differ in their nucleosome positioning pattern around gene ends. We first analyzed two promoter types, namely, active and poised promoters, defined using TC-rMNase-seq [28]. By extracting and averaging the nucleosome occupancy from the experimental MNase-seq nucleosome map of mouse ESCs, we found that, compared with poised genes, active genes had a stronger phasing of nucleosomes around the TSS (Figure 1A). Particularly, active genes displayed a stronger +1 nucleosome signal than bivalent genes. Additionally, active genes also displayed a higher level of nucleosome occupancies surrounding the nucleosome depleted regions (NDR) at TTS (Figure 1A).

The chemical map approach, however, indicated that active genes had much higher levels of nucleosome occupancies around both the TSS and TTS, and the nucleosome depletion observed in the MNase-seq map at the TTS, disappeared (Figure 1B). The part of nucleosome depleted region located immediately upstream of the TSS, observed in the MNase-seq map, was enriched with more nucleosomes in the chemical map (Figure 1B). As previously discussed by others [10], the discrepancy between the MNase-seq map and the chemical map might have been caused by the enrichment of fragile nucleosomes at the TSS and TTS, which are sensitive to MNase digestion. It is noteworthy that a high level of nucleosome occupancy does not necessarily mean increased compaction of chromatin [13]. Consistent with this, a high level of nucleosome occupancy was observed in the chemical map at the borders of active genes (Figure 1B), which were suggested previously to be located in more accessible regions at the chromatin compaction level [28].

We further analyzed nucleosome occupancy profiles for three types of promoters defined according to histone marks [26]. For genes with activating histone mark (H3K4me3) at their promoter regions, nucleosome depletion at the TSS was more pronounced than the other two types (H3K27me3 and bivalent) in MNase-seq mapping (Figure 1A, third plot). For genes carrying the repressive mark (H3K27me3) at their promoter regions, the global nucleosome occupancy level was much higher around the TSS than that of the H3K4me3-positive and bivalent genes, and there was no NDR at the TSS of the H3K27me3-positive genes (Figure 1A, third plot). By contrast, chemical mapping showed that H3K4me3-positive promoters, compared with the other two types of promoters, were occupied by a remarkably higher level of nucleosomes even at the immediate upstream of the TSS that was depleted of nucleosomes in the MNase-seq map (Figure 1B, third plot). It was also clear that H3K4me3-positive genes have phased nucleosomes upstream of their TSS. These results, overall, coincided with the corresponding results for the active and poised promoters. Contrasting with active and poised promoters, the nucleosome occupancy profile around repressive promoters carrying the H3K27me3 mark was flat (Figure 1B).

### 2.2. Sequence-Dependent Nucleosome Positioning around Promoters with Distinct Chromatin States in ESCs

As shown in the chemical mapping above, the genes within distinct chromatin states have characteristic nucleosome positioning patterns. For example, active genes have higher levels of nucleosome occupancy at gene ends than repressed and poised genes, and nucleosomes are enriched at the TTS despite chromatin states. We then considered to what extent DNA intrinsic properties could explain the nucleosome organization at gene ends. The sequence-dependent deformation energy model (model #1, see methods) reproduced the nucleosome depletion at the TTS as depicted in the MNase-seq map (Figure 2), but failed to predict nucleosome occupancy around the TSS. The predicted nucleosome organization was in an even more remarkable disagreement with the chemical map. These results suggest that our deformation-energy-based nucleosome occupancy model (model #1) is not sufficiently powerful to predict nucleosome occupancy in mouse ESCs, despite its successful application in the yeast genome [21]. It is, therefore, necessary to develop other models (see below).

Before we discuss an improved model for nucleosome occupancy prediction, we must first explore another aspect of the nucleosome: rotational positioning. Although our deformation-energy-based nucleosome occupancy model (model #1) had poor performance in nucleosome occupancy prediction in mouse ESCs, it was confirmed that rotational positioning of nucleosomes in budding yeast as well as mouse could be accurately predicted with the model [21,24]. DNA bending is an ideal predictor of the rotational positioning of nucleosomes [21,22,24]. Based on the original bending energy calculation, we developed an improved version of nucleosome rotational positioning index by taking the correlation between bending energy profile and wave packet into account (see Methods section). The nucleosome rotational positioning index predicted 71% of the rotational setting of unique nucleosomes in mouse ESCs with a prediction error ≤2 bp (Figure 3A). A successful prediction meant that the DNA major groove side at the nucleosome dyad faced the histone octamer, which was consistent with the experimental observation. Although the improvement in the prediction performance was subtle when compared with that of raw bending energy (71% vs. 69%), the contour of the rotational positioning index landscape was able to provide a much stronger signal about the translational position of nucleosomes (Figure 3B).

To further explore the rotational positioning of nucleosomes around the TSS, we compared the bending energies that corresponded to three representative regions located upstream, central, and downstream of the TSS. The results showed that the upstream region of the TSS had the strongest 10-bp oscillation in bending energy, while the central region at TSS had the weakest oscillation (Figure 4). The amplitude of 10-bp oscillation in bending energy did not appear to differ substantially between the active genes and poised genes. According to our model, the 10-bp periodical oscillation of bending energy was a strong indicator of bending anisotropy, thereby suggesting the rotational positioning signal. Our results indicated that, compared with the downstream regions of TSS, the gene upstream regions preferentially deposited rotationally locked nucleosomes. We also observed that the nucleosomes at the central regions of TSS did not necessarily have local minima of bending energies at their center positions (Figure 4).

Additional analyses of rotational positioning strength by using both standard deviation of bending energies and 10-bp periodical signal estimated by fast Fourier transform revealed V-shaped patterns in rotational positioning signal immediately downstream of the TSS (Figure 5A,B). The V-shape-like pattern, as well as the enhanced rotational positioning signal at the TTS (Figure 5A,B), resembled the nucleosome organization discovered in the chemical map (Figure 1B), suggesting that rotational positioning strength encoded in DNA might be an important factor determining the nucleosome positioning pattern observed in the chemical map. Indeed, analyzing the rotational positioning index (RPI)-based nucleosome occupancy (model #2), we produced similar nucleosome positioning patterns around the TSS and TTS, as in the chemical map (Figure 5C). Based on Figure 5, we also proposed that the nucleosome positioning information around both ends of genes was largely encoded in the DNA sequence, and similarly between active (H3K4me3-positive) and poised (bivalent) genes. However, our model failed to predict the nucleosome phasing around the TSS of active genes (compare Figure 5C and Figure 1B). The DNA intrinsic code for positioning nucleosomes around the TSS for repressive genes greatly differed from the other two types.

### 2.3. Nucleosome Organization around Promoters with Different CpG Composition in ESCs

Promoters with different CpG composition are functionally different. It was shown that CpG-rich promoters enriched with H3K4me3 mark tend to regulate house-keeping genes and genes expressed during embryonic development, while CpG-poor promoters regulate tissue-specific genes [26]. This difference is usually coupled with epigenetic marks, whose change (gain or loss) between ESCs and differentiated cells directs corresponding gene expression. For example, some of the CpG-rich promoters with bivalent marks in ESCs have lost one type of histone mark in differentiated cells, suggesting the gain of either an activating or repressive role during cell differentiation [26]. The majority of CpG-poor promoters in ESCs lack both H3K4me3 and H3K27me3 marks. How do the promoters with different CpG composition differ in terms of nucleosome positioning? It was revealed in an MNase map that HCG promoters (CpG-rich promoters) were mostly depleted of nucleosomes, whereas LCG and ICG promoters had a higher level of nucleosome occupancy (Figure 6A). Chemical mapping also confirmed the higher level of nucleosome occupancy for LCG and ICG promoters, but only downstream of the TSS (Figure 6B). Compared with the HGC group, our RPI-based nucleosome occupancy prediction also suggested a global enrichment of nucleosomes around the TSS for LCG and ICG groups (Figure 6C). The nucleosome phasing around the TSS for HCG genes could not be predicted with our model (Figure 6C). Because the majority of HCG promoters were marked with H3K4me3, and, thus, our results implied that nucleosome phasing near the TSS of HCG genes might be largely directed by external factors such as remodelers rather than sequence preference. It was interesting that the nucleosome phasing signal downstream of the TSS of LCG genes was somewhat reproducible with our sequence-dependent model (Figure 6C). It is possible that the nucleosomes positioned around CpG-poor promoters are not susceptible to remodelers because of the lack of histone marks, thereby highlighting the sequence-directed phasing.

### 2.4. Nucleosome Organization around Transcription Factor-Binding Sites in ESCs

Nucleosomes were arranged differentially around transcription factor-binding sites. Some transcription factors, such as Zxf, Klf4 and c-Myc, were shown to preferentially bind to nucleosome-depleted regions, while others, including pluripotency transition factors such as Oct4, Sox2 and Nanog tended to target nucleosomes [27], acting as pioneer TFs in chromatin remodeling during cell reprogramming [29]. Chemical mapping also discovered strong nucleosome phasing around TF binding sites [10], but whether these positioning patterns are directed by DNA sequence preference, demands further investigation. By analyzing the DNA sequences of the transcription factor-binding sites with our model, we found that the patterns obtained from RPI-based nucleosome occupancy calculation were in good agreement with experimental results (Figure 7). Our model successfully estimated the nucleosome depletion at the binding sites of Klf4, c-Myc and Zxf, as well as the nucleosome peaks at the centers of binding sites of Oct4, Sox2, and Nanog. Consistent with experimental results [27], the binding sites of Smad1 were predicted to be enriched with nucleosomes in a broader range than Oct4, Sox2 and Nanog, which were characterized with a narrow sharp peak in nucleosome occupancy (Figure 7). Our model not only revealed sequence-dependent nucleosome enrichment at the central binding sites of Oct4, Sox2, and Nanog, but also produced some peaks in predicted nucleosome occupancy (Figure 7) that were consistent with the nucleosome phasing detected in the chemical map [10]. It was noteworthy that Klf4 binding sites were deficit in nucleosomes in both the MNase-seq map and prediction, but were enriched with nucleosomes in the chemical map (Figure 7), suggesting that nucleosome positioning at the binding sites of Klf4 in ESC is not mediated by DNA sequence. In addition, CTCF was involved in the establishment of 3D genomic architecture by binding to TAD boundaries and loop anchor sites [30], and CTCF binding sites were predicted to be enriched with nucleosomes (Figure 7), supporting the notion that CTCF and nucleosomes are preferentially co-occupied in ESCs [10]. Again, this is contrasted with the nucleosome depletion at the center of CTCF-binding sites observed in the MNase-seq data [27].

### 2.5. Conclusions

Our results imply that the rotational positioning signal encoded in the DNA sequence is highly predictive of chemical-mapping-determined nucleosome positioning patterns. The rotational positioning signal is captured in the deformation energy calculation, and, furthermore, strengthened by calculating the inner product (such as cross-correlation) between the bending energy profile and wave packet function. This might be responsible for its successful application presented in this study. A deep-learning approach termed DNAcycP was developed for intrinsic DNA cyclizability prediction [31]. DNA-directed nucleosome positioning was also correlated with DNA cyclization probability [31]. DNAcycP-based analyses revealed troughs of DNA cyclizability at the TSS and TTS (Figure 8), which were consistent with the nucleosome occupancy pattern observed in the MNase-seq map, but not with that in the chemical map (Figure 1). This indicates that our bending energy-based model (model #2) outperforms DNAcycP in capturing chemical-map-based nucleosome positioning characteristics.

The previous studies showed that the nucleosome occupancy landscape around the gene ends displayed great discrepancy between the MNase-seq map and the chemical map for mouse ESCs [10]. The discrepancy was likely to be caused by the inaccurate mapping of MNase-sensitive nucleosomes because of sequence bias in MNase cleavage. Assuming the chemical mapping approach is able to give a more accurate determination of nucleosome positions in vivo, we provided for the first time, to our best knowledge, computational supporting evidence for the sequence-dependent properties of the nucleosome positioning pattern in mouse ESCs. Our results further provided informative knowledge about if, and how, the distinct chromatin states in ESCs differ in terms of sequence dependence. Our results suggested that the MNase-sensitive nucleosomes enriched around the TSS and TTS are likely to be encoded in the DNA sequence. Rotational positioning-associated properties reflected in DNA deformability were suggested as being the major factor affecting nucleosome patterns around both ends of genes, but also at binding sites of transcription factors that have key regulatory functions in ESCs. Given the remarkable difference in sequence dependence between the distinct chromatin states (e.g., active and repressive) and the switch between chromatin states that would take place during cell differentiation, it can be proposed that basic information about nucleosome positioning in ESCs is encoded in the DNA sequence, and epigenetic change dominates the change of the chromatin state.

As shown in the present study, the success of bending energy in the inference of nucleosome positioning highlights the role of the bending property of DNA in nucleosome positioning in mouse ESCs. Note that it is unclear why the original shearing energy-based nucleosome occupancy model that achieved a good performance in budding yeast [21] was unable to predict the chemical-map-based nucleosome occupancy in mouse ESCs. As our deformation energy model is independent of sequence bias that may exist in training-based computational tools due to MNase cleavage bias, it may provide a more unbiased inference about pure DNA-dependent properties of nucleosomes. In addition, one can expect that the rotational positioning index defined on the bending energy may be used as a quality check tool for evaluating the quality of experimentally determined nucleosome center positions or inferred rotational positioning. Moreover, because the binding of transcription factors and RNA polymerase II to DNA depends on rotational positioning [32,33], and analogously, the binding of developmentally important transcription factors (e.g., pioneer transcription factors) to nucleosomal DNA is probably modulated strongly by the rotational setting of nucleosomes, the rotational positioning index may provide a deeper insight into TF binding affinity and its role in specific cell types. In the future, it will be interesting to see for which types of cells and which subset of genes during embryo development and cell differentiation, that nucleosome positioning is determined largely by DNA sequence rather than external factors. 

## 3. Materials and Methods

### 3.1. Genes with Distinct Chromatin States

We used two datasets of chromatin states in mouse ESCs. One was defined by using a time-course digestion of chromatin using moderate MNase levels (TC-rMNase-seq). Based on TC-rMNase-seq, Yu et al. 2020 defined two types of local chromatin structure: one was open chromatin characterized with high transcription activity and active histone modification marks such as H3K4me3 and H3K27ac, whereas the other one was suggested to be moderately condensed regions enriched with both active and repressive histone marks corresponding to the poised (or bivalent) chromatin state [28]. The gene IDs whose promoter regions overlapped with the above two regions were provided in their study [28]. We obtained the positional information (mm9-based) of the transcription start sites (TSS) and transcription termination sites (TTS) of the genes by mapping the gene IDs to the refGene list downloaded from UCSC, and retaining only the unique genomic positions. The final data included 1056 genes without bivalent histone marks (H3K4me3 and H3K27me3) and 1341 genes with bivalent histone marks. The other dataset consisted of three types of promoters in mouse ESCs, which were defined according to the chromatin modification marks (H3K4me3, H3K27me3, bivalent) in Mikkelsen et al. 2007 [26]. Classification of promoters according to CpG content was also obtained from Mikkelsen et al. 2007 [26]. The mm8-based coordinates of the promoters were converted to mm9 using LiftOver tool (http://genome.ucsc.edu/cgi-bin/hgLiftOver, accessed on 1 July 2022).

In addition, ChIP-seq determined binding sites of transcription factors, such as Oct4, Sox2, Nanog Zxf, Klf4 and c-Myc were downloaded from Chen et al. 2008 [34].

### 3.2. Experimental Nucleosome Maps of Mouse ESCs

Two nucleosome maps of mouse ESCs were taken from the previous study [10] and used to show the nucleosome positioning patterns around two ends of genes. One was obtained with a complete MNase-seq in wild type ESCs, and the other was obtained with a chemical cleavage method that was based on the cleavage of nucleosome DNA utilizing Fenton reaction in H4S47C mutant ESCs. Center-weighted nucleosome occupancy maps were obtained exactly following their methods [10]. We called a redundant nucleosome map from the raw nucleosome center positioning (NCP) scores using R package NuCMap [15,35]. In order not to exclude many possible nucleosome positions overlapping within a narrow region, we called the redundant nucleosome map allowing the centers of two adjacent nucleosomes to be away from each other by at least 10 bp.

### 3.3. DNA Deformation-Energy-Based Descriptors

We used our previously developed DNA deformation energy model [21] to investigate sequence-dependent properties of chromatin states in mouse ESCs. Briefly, the model calculated two forms of deformation energies, bending energy and shearing energy, for any 129-bp DNA segment expected to wrap around a histone octamer. DNA deformation was described according to the base-pair step model [36]. Base-pair step parameters for a DNA segment expected to form a nucleosome were estimated using two structural constraints (e.g., global curvature and pitch) derived from crystal structures of nucleosomal DNA [37]. The details of the model are available in the original article [21].

Based on the deformation energy, several descriptors were defined as follows. First, nucleosome occupancy (referred to as model #1) at genomic position i was estimated as
(1)$$\;\mathrm{occ}=\sum_{k=-73}^{73}w_{k}p_{i+k}$$
where $p_{i}=e^{-\beta E_{i}}$ is nucleosome positioning score at position i, $E_{i}$ is the deformation energy (bending energy or shearing energy) normalized in the range [–1,1] by using genome-scale max and min values, and $w_{k}=e^{-\frac{1}{2}{\left(\frac{k}{20}\right)}^{2}}$ is Gaussian weight as defined in Voong et al. 2016 [10]. β was set to 1 for computational simplicity. According to our previous study [21], shearing energy is powerful in the nucleosome occupancy estimation.

In the deformation energy calculation, the 129-bp sliding window would exceed the length of the shorter inter-nucleosome linkers and consequently may cause failure in the detecting narrow NDRs due to contamination of nucleosomal DNA in the sliding window, when the calculation is performed along the chromosome. To overcome this, we tested several other window sizes (75 bp, 101 bp and 129 bp) and chose 101 bp in this study. Sliding windows much shorter than this one are not suggested, due to their high fluctuation effect in deformation energy calculation. We will see below, that the 101-bp window size is also suitable for another nucleosome occupancy index that is defined on the bending energy. Throughout this study, the unit of deformation energy is kT/bps, where k is Boltzmann constant, T is effective temperature, and bps denotes the base-pair step. In other words, the deformation energy is normalized by the count of base-pair steps in the 101-bp window.

It was previously shown that shearing energy is powerful in the nucleosome occupancy estimation, and bending energy is a good predictor of rotational positioning [21]. In this study, we also defined bending-energy-based nucleosome occupancy index (referred to as model #2) and confirmed its performance. Bending energy calculated for a 101-bp genomic segment was assigned to its central nucleotide and the inner product between the vector of 147 consecutive nucleosome positioning scores ($p_{i}=e^{-\beta E_{i}}$) and the wave packet function $e^{-\frac{k^{2}}{800}}\times e^{99.906ki}$ (complex number) was computed to strengthen the oscillation signal of bending energy at the nucleosomal central region. The *k* integer adopted was from −73 to 73. The wave packet function had 10-bp periodicity with the peak value at the center of the wave packet (Figure 9A). This inner product actually behaves as a cross-variance between two vectors and is termed ‘rotational positioning index’ because of its ability to indicate the rotational setting of nucleosomes. The higher the index is, the more likely the nucleosome center is to be placed at that genomic position. Therefore, the value of the rotational positioning index is also called the nucleosome center score. After the rotational positioning index was calculated along a genomic sequence, the upper contour of the rotational positioning index was subjected to an envelope fitting. The fitted value at each genomic site was our second method of estimating nucleosome occupancy, which was solely defined on bending energy. The calculation steps are illustrated in Figure 9B.

In addition, to depict the rotational positioning property of nucleosomes, two other indices were defined. Rotational positioning strength was defined as the standard deviation of the bending energies within a 147-bp sliding window along the DNA sequence. According to our model [21], for a nucleosome, the major groove side of the DNA at the position with a local bending energy minimum, preferentially faced the histone octamer. Moreover, the bending energies displayed a strong 10-bp periodical oscillation pattern along the nucleosomal DNA sequence. Therefore, the standard deviation of bending energies within a 147-bp region can be used as a strong indicator of the rotational positioning strength of a nucleosome. Another useful indicator of rotational positioning strength is the amplitude of the fast Fourier transform of bending energies.

## Figures and Tables

**Figure 1 ijms-23-14488-f001:**
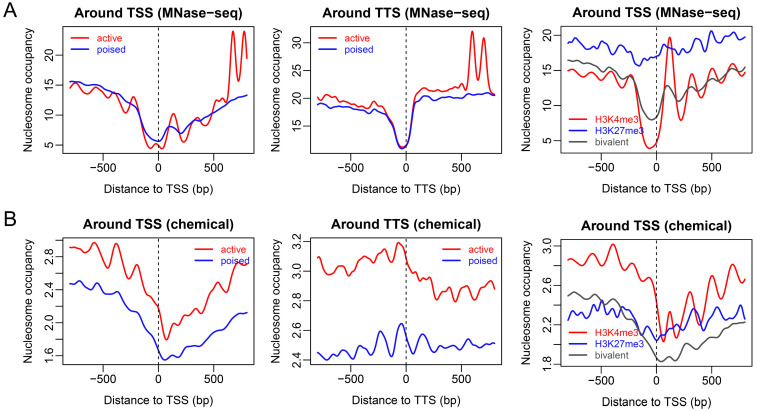
Experimental nucleosome organization around the TSS and TTS for genes in distinct chromatin states. (**A**) MNase-seq based nucleosome occupancy. (**B**) Chemical-mapping-based nucleosome occupancy. In the plots, active and poised promoters were defined in Yu et al. 2020 [28], and the promoters carrying different histone marks (H3K4me3, H3K27me3, bivalent) were defined in Mikkelsen et al. 2007 [26].

**Figure 2 ijms-23-14488-f002:**
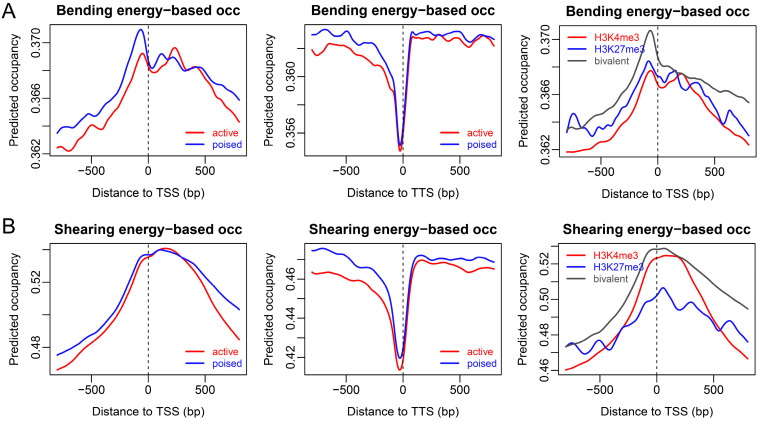
Nucleosome occupancy predicted with our deformation energy model (model #1) at the TSS and TTS for genes in distinct chromatin states. (**A**) Bending energy-based prediction. (**B**) Shearing energy-based prediction. The legends for chromatin states are the same as in Figure 1.

**Figure 3 ijms-23-14488-f003:**
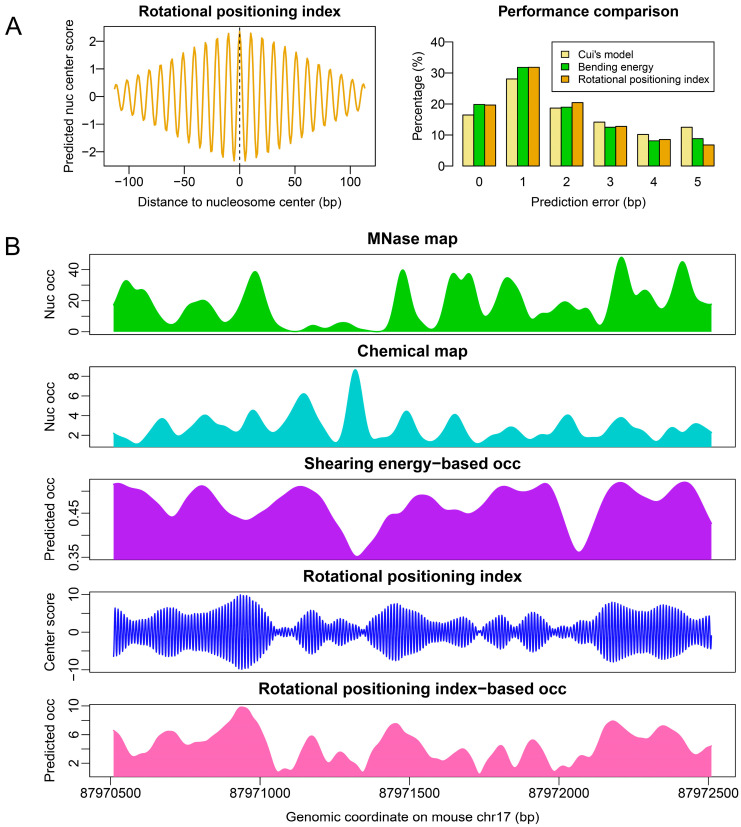
Rotational positioning index defined on DNA bending energy is capable of predicting rotational setting and translational positions of nucleosomes in mouse. (**A**) Testing on the unique nucleosomes with high center-positioning scores in mouse ESCs, the peak of rotational positioning index coincides with the nucleosome center positions. The prediction error denotes the distance between the experimental nucleosome position and the position with the highest nucleosome center score (rotational positioning index) in the interval [−5, +5] around the real nucleosome position. The ordinate denotes the percentage of error in the total tested nucleosome positions. (**B**) Rotational positioning index-based occupancy (obtained with model #2) coincides with the experimental occupancy, even with the nucleosomes occupied in the chemical map but not in the MNase map. The shearing energy-based occupancy landscape (obtained with model #1) is also provided.

**Figure 4 ijms-23-14488-f004:**
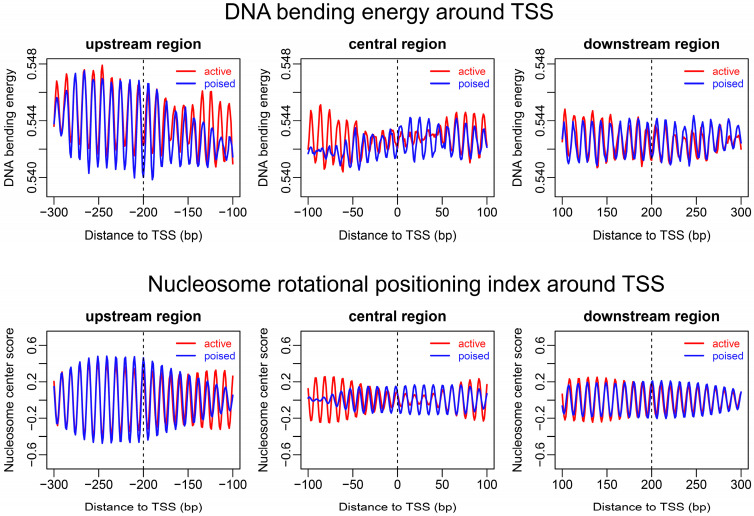
DNA bending energy and nucleosome rotational positioning index around the TSS of active and poised genes. The dash line in each plot indicates the genomic site around which nucleosome centers in the redundant map were called in a 50-bp span and aligned. The unit of bending energy is kT/bps, where k is Boltzmann constant, T is effective temperature, and bps denotes base-pair step.

**Figure 5 ijms-23-14488-f005:**
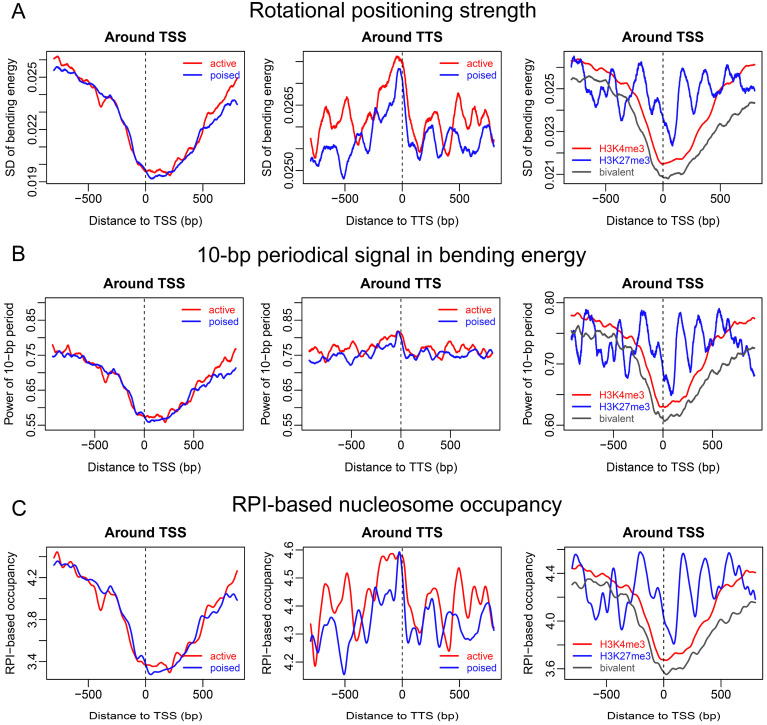
Rotational positioning signal and estimated nucleosome occupancy inferred from DNA bending energy around two ends of genes in distinct chromatin states. (**A**) Rotational positioning strength was estimated by the standard deviation of bending energy. (**B**) Ten-periodical signal in bending energy was also directly measured by the power of fast Fourier transform at the period of 10 bp. (**C**) Rotational positioning index (RPI)-based nucleosome occupancy displayed a similar trend as in the chemical map. A 50-bp sliding window was used in the calculation of both rotational positioning strength and ten-periodical signal.

**Figure 6 ijms-23-14488-f006:**
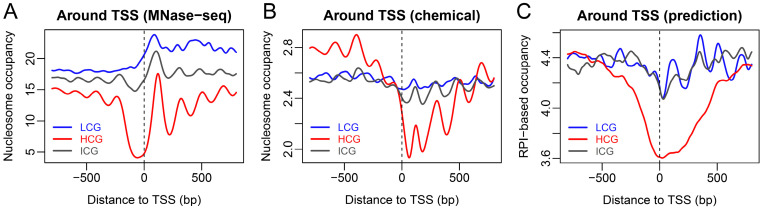
Nucleosome organization around the TSS for the promoters with different CpG composition defined in Mikkelsen et al. 2007 [26]. (**A**) MNase-seq determined nucleosome occupancy. (**B**) Chemical mapping determined nucleosome occupancy. (**C**) RPI-based nucleosome occupancy prediction. Promoters with low (LCG), high (HCG), or intermediate CpG composition (ICG) were separately analyzed.

**Figure 7 ijms-23-14488-f007:**
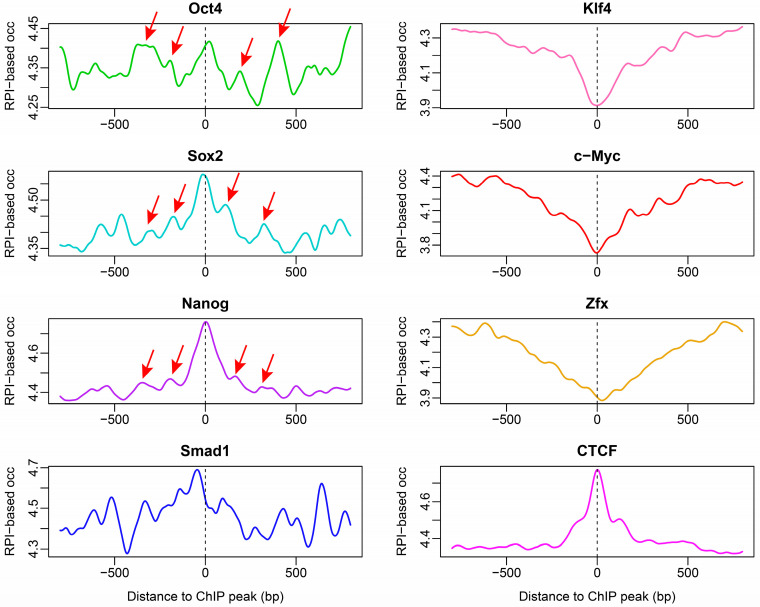
Rotational positioning index (RPI)-based prediction of nucleosome occupancy around the binding sites of transcription factors. The red arrows suggest, although weakly, the nucleosomes phased around the central peak.

**Figure 8 ijms-23-14488-f008:**
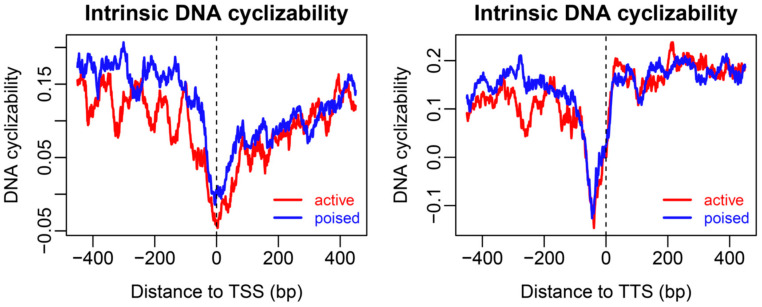
Intrinsic DNA cyclizability predicted with a deep-learning based approach, DNAcycP [31]. The legends are the same as in Figure 1.

**Figure 9 ijms-23-14488-f009:**
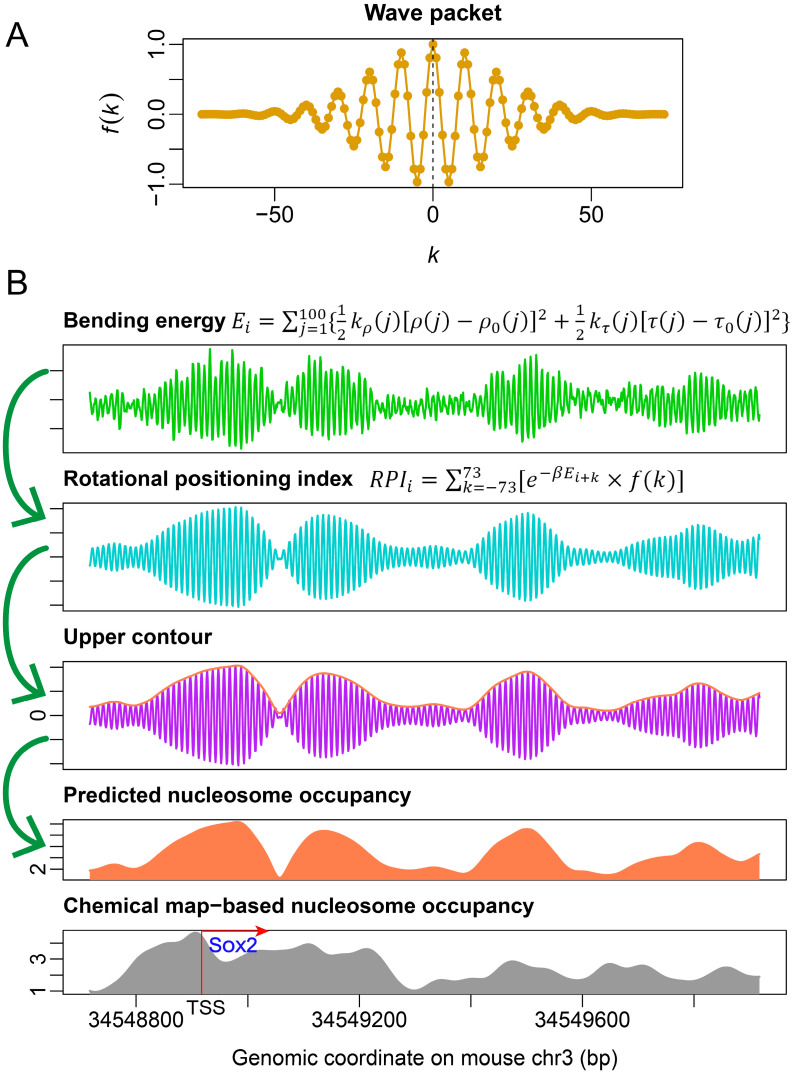
Schematic diagram for bending-energy-based nucleosome occupancy calculation. (**A**) Plot of the wave packet function $f\left(k\right)=e^{-\frac{k^{2}}{800}}\times e^{99.906ki}$ with k∈[−73,73]. (**B**) Schematic diagram for nucleosome occupancy calculation for a genomic region around gene Sox2. j∈[1100] corresponds to the base-pair steps in a 101-bp window used in the bending energy calculation. For comparison purposes, the chemical-map-based nucleosome occupancy was also provided.

## Data Availability

Experimental nucleosome positioning data including chemical map and MNase-seq map were downloaded from GEO (GSE82127) and ChIP-seq data for transcription factors were downloaded from GEO (GSE11431).

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
