# Peer review of "DNA Sequence-Dependent Properties of Nucleosome Positioning in Regions of Distinct Chromatin States in Mouse Embryonic Stem Cells"

_ijms, 2022, doi:10.3390/ijms232214488_

Round 1

Reviewer 1 Report

The authors have investigated the “DNA sequence-dependent properties of nucleosome positioning in regions of distinct chromatin states in mouse embryonic stem cells” using DNA deformation energy model. The work seems interesting. However, I have some major concerns before going ahead to accept it. 

The abstract and the introduction are well written. The results somehow seem to be a combination of results and discussion while the discussion seems more like a conclusion. Please rewrite or change the heading to make it more appropriate. If the heading is “results” try not to include discussions part on it and save it for the actual discussion part. 

Include one para to elaborate on how this work can benefit the scientific world. As the model is not universal and needs modification based on the source of genome as mentioned in line 182, a convincing explanation on the gravity of the work is needed.

 Abbreviations, no matter how generic, must be elaborated at the very first mention.

If the authors make improvements based on these suggestions and convince that in spite of the shortcomings of the model, the work can contribute to the scientific community then the manuscript can be accepted.

Author Response

We are grateful to the reviewers for their valuable comments. We revised carefully the manuscript. In addition to the corrections requested by reviewers, we also corrected the legends in Figure 6 and replaced Figure 9 to make our model more understandable. All the corrections are marked in red in the revision. Our responses to reviews are listed below.

Reviewer 1:

The authors have investigated the “DNA sequence-dependent properties of nucleosome positioning in regions of distinct chromatin states in mouse embryonic stem cells” using DNA deformation energy model. The work seems interesting. However, I have some major concerns before going ahead to accept it. 

  1. The abstract and the introduction are well written. The results somehow seem to be a combination of results and discussion while the discussion seems more like a conclusion. Please rewrite or change the heading to make it more appropriate. If the heading is “results” try not to include discussions part on it and save it for the actual discussion part.

Response: Thank you for the comments. We tried to change the heading of “Results” into “Results and discussion” and the “Discussion” into “Conclusions” as a part of “Results and discussion”. However, we found that after the inclusion of discussion about our model and its possible applications, it is somewhat inappropriate to place the content in the conclusion part. So, we retained the original structure of the manuscript.

  1. Include one paragraph to elaborate on how this work can benefit the scientific world. As the model is not universal and needs modification based on the source of genome as mentioned in line 182, a convincing explanation on the gravity of the work is needed.

Response: we appreciate the comments. We supplemented a paragraph as follows: As shown in the present study, the success of bending energy in the inference of nucleosome positioning highlights the role of bending property of DNA in nucleosome positioning in mouse. Note that it is unclear why the original shearing energy-based nucleosome occupancy model that achieved a good performance in budding yeast (Liu et al. 2016) was unable to predict chemical map-based nucleosome occupancy in mouse ESCs. Because our deformation energy model is independent of sequence bias that may exist in training-based computational tools due to MNase cleavage bias, it may provide a more unbiased inference about pure DNA-dependent properties of nucleosomes. In addition, one can expect that the rotational positioning index defined on the bending energy may be used as a quality check tool for evaluating the quality of experimentally determined nucleosome center positions or inferred rotational positioning. Moreover, the binding of developmentally important transcription factors (e.g. pioneer transcription factors) to nucleosomal DNA is probably modulated strongly by the rotational setting of nucleosomes, and thus the rotational positioning index may provide us a deeper insight into the TF binding affinity and its role in specific cell types. It is interesting to see in future that for what types of cells and which subset of genes during embryo development and cell differentiation the nucleosome positioning is determined largely by DNA sequence rather than external factors.

  1. Abbreviations, no matter how generic, must be elaborated at the very first mention.

Response: thank you for the comments. We revised as suggested.

  1. If the authors make improvements based on these suggestions and convince that in spite of the shortcomings of the model, the work can contribute to the scientific community then the manuscript can be accepted.

Response: thank you. We revised the manuscript according to the comments.

Reviewer 2 Report

The authors of the article "DNA sequence-dependent properties of nucleosome positioning in regions of distinct chromatin states in mouse embryonic stem cells" devoted their work to an important and relevant topic. We investigated the sequence-dependent organization of nucleosomes within chromatin states in mouse embryonic stem cells. The authors use original calculation methods based on the strain energy. As a result, the authors identified sequence-dependent properties of the location of nucleosomes in regions of different chromatin states in mouse embryonic stem cells.

There are a few formal remarks to the work:

1. In the article, references to literature are indicated in brackets with the author's name and year, it is desirable to remake references to sources in the form of continuous numbering of references in the text.

2. In almost all the figures, part of the text looks strange - somewhere there are narrow lines, somewhere the text is slightly cut off at the top.

3. Figure 4 does not indicate the units of DNA bending energy

4. In figure 10, the units of measurement along the axes are also not signed.

Author Response

We are grateful to the reviewers for their valuable comments. We revised carefully the manuscript. In addition to the corrections requested by reviewers, we also corrected the legends in Figure 6 and replaced Figure 9 to make our model more understandable. All the corrections are marked in red in the revision. Our responses to reviews are listed below.

Reviewer 2:

The authors of the article "DNA sequence-dependent properties of nucleosome positioning in regions of distinct chromatin states in mouse embryonic stem cells" devoted their work to an important and relevant topic. They investigated the sequence-dependent organization of nucleosomes within chromatin states in mouse embryonic stem cells. The authors use original calculation methods based on the strain energy. As a result, the authors identified sequence-dependent properties of the location of nucleosomes in regions of different chromatin states in mouse embryonic stem cells. There are a few formal remarks to the work:

  1. In the article, references to literature are indicated in brackets with the author's name and year. It is desirable to remake references to sources in the form of continuous numbering of references in the text.

Response: thank you for the comments. We revised as suggested.

  1. In almost all the figures, part of the text looks strange - somewhere there are narrow lines, somewhere the text is slightly cut off at the top.

Response: the figures embedded in the main text were snapshots of original figures in pdf format. The figures appear normal on my computer. We provided the figures with high resolution in pdf format. If it still seems abnormal, please let me know.

  1. Figure 4 does not indicate the units of DNA bending energy

Response: thank you for the comment. Throughout this study, the unit of deformation energy is kT/bps, where k is Boltzmann constant, T is effective temperature, and bps denotes base-pair step. In other words, the deformation energy is normalized by the count of base-pair steps in the 101-bp window. This has been clarified in the method section and in the Figure caption.

  1. In figure 9, the units of measurement along the axes are also not signed.

Response: we would like to clarify that the variables along the both axes should not have units.

Round 2

Reviewer 1 Report

The authors tried to incorporate the changes as suggested. However, for the sake of the readers and considering the widely accepted format of scientific writing, I will strongly suggest including a short conclusion section. Rest seems fine to me and If this is taken care of I suggest accepting the manuscript.

Author Response

Thank you for the suggestion. We changed the heading of “Results” into “Results and discussion” and the “Discussion” into “Conclusions” as a part of “Results and discussion”.